# Novel Insights into T-Cell Exhaustion and Cancer Biomarkers in PDAC Using ScRNA-Seq

**DOI:** 10.3390/biology14081015

**Published:** 2025-08-07

**Authors:** Muhammad Usman Saleem, Hammad Ali Sajid, Muhammad Waqar Arshad, Alejandro Omar Rivera Torres, Muhammad Imran Shabbir, Sunil Kumar Rai

**Affiliations:** 1Department of Biological Sciences, Faculty of Sciences, International Islamic University, Islamabad 04436, Pakistan; muhummad.bsbt1296@iiu.edu.pk (M.U.S.); hammad.bsbt1290@iiu.edu.pk (H.A.S.); 2Department of Molecular Biology, Shaheed Zulfiqar Ali Bhutto Medical University, Islamabad 44080, Pakistan; waqararshad388@gmail.com; 3Department of Molecular Sciences, University of Medicine and Health Sciences, Basseterre KN 01018, Saint Kitts and Nevis; alejrivera@umhs-sk.net

**Keywords:** PDAC, T-cell exhaustion, immune landscape of PDAC, PDAC time, PDAC heterogeneity

## Abstract

Pancreatic ductal adenocarcinoma (PDAC) is a very aggressive and deadly cancer that is hard to treat due to the hostile environment created by the cancer cells, which exhausts the immune system, especially T-cells, which are important cells that usually help the body fight cancer. In this study, we used single-cell RNA sequencing (scRNA-seq) to examine individual cells from PDAC tumors and healthy tissues and identify which genes were unusually active in cancer cells and T-cells. We found specific genes altered in cancer by comparing cancer cells to normal cells. We also studied T-cells to find genes that may be exhausted by signals from the cancer cells. This helped us understand how cancer cells might be disturbing T-cell function, leading to T-cell exhaustion and making it harder for the immune system to fight the tumor. This research identified crucial genes that are unusually active in cancer cells and T-cells, resulting in T-cell exhaustion. We then confirmed these results using public databases and checked how these genes relate to patient survival. This research discovered new genetic markers that might be important in how PDAC cancer cells escape the immune system, offering possible new targets for future therapies.

## 1. Introduction

Pancreatic tumors are classified as epithelial or non-epithelial based on their histogenesis, where pancreatic ductal adenocarcinoma (PDAC) is the most common malignant epithelial tumor of the pancreas, accounting for more than 85% of all pancreatic malignancies [1]. The cancerous exocrine duct cells lining the pancreas lead to PDAC and are the deadliest of all adult abdominal tumors [2]. PDAC is exceptionally lethal due to its typically late diagnosis, aggressive metastatic behavior, and profoundly desmoplastic and hypovascular tumor microenvironment [3,4]. Moreover, PDAC has many subtypes, including classical and basal, squamoid-basaloid, treatment-enriched, and quasi-mesenchymal [5,6].

Furthermore, PDAC is the third-leading cause of cancer death in the age groups 50–64 and 65–79; meanwhile, there is only a 10.8% 5-year overall survival rate for PDAC for both metastatic and resectable cases in the United States (US) [7,8]. Strikingly, 62,210 cases were reported in the US in 2023 and 49,380 deaths were reported in 2022, while PDAC accounts for 2% of all cancer cases and results in 5% of all cancer deaths in the US, therefore underlining the crucial need for earlier detection [7,9]. Notably, PDAC is estimated to become the second-leading cause of cancer deaths by 2030, surpassing breast cancer, as mortality rates are on the rise, increasing 1% annually [10]

The major risk factors of PDAC include smoking, obesity, diabetes, a family history of pancreatic cancer, and inherited genetic mutations such as those in *BRCA1*, *BRCA2*, and *PALB2* [11,12]. Subsequently, the common symptoms of PDAC patients include jaundice due to bile duct obstruction, unexplained weight loss, abdominal or back pain, and new-onset diabetes. These symptoms typically arise at advanced stages, contributing to the poor prognosis associated with PDAC [13].

Currently, different treatment regimens, including surgery, chemotherapy, and radiation therapy, are used to treat PDAC; however, these treatment options remain ineffective due to highly resistant PDAC [14]. Moreover, the gold standard to treat PDAC is polychemotherapy regimens, such as gemcitabine and abraxane or FOLFIRINOX [15].

Recently, advances in immunotherapy have led to a shift in treatment regimens by gaining insights into the intricate interactions between the immune system and cancer cells. At the same time, PDAC has been resistant to immunotherapy and has demonstrated marginal efficacy in terms of survival [16,17]. Immunotherapy faces hurdles in PDAC treatment due to the immunosuppressive microenvironment, which hinders T-cell infiltration and activation, limiting the effectiveness of the combination therapy [18]. The complex tumor immune microenvironment (TIME) of PDAC modulates the infiltration of the immunosuppressive cells and the activity of immune regulatory molecules, leading to the anti-tumor immune responses’ dysfunctionality, including the T-cells’ exhaustion [19].

T-cell exhaustion is the dysfunctional state of the T-cells in chronic infections and cancer that results in cancer immune evasion, eventually leading to the poor prognosis associated with many cancers [20]. In PDAC, cancer cells release tumor-associated antigens (TAAs) into the tumor immune microenvironment (TIME), where antigen-presenting cells (APCs) such as dendritic cells, macrophages, and antigen-presenting cancer-associated fibroblasts (apCAFs) internalize and present these antigens via major histocompatibility complex (MHC) class II molecules to CD4^+^ T-cells. This repeated MHC-II–TCR engagement triggers chronic T-cell receptor (TCR) stimulation, causing T-cell hyperactivation. Within the harsh PDAC milieu, elevated levels of cytokines such as IL-6 and TGF-β exacerbate metabolic stress in T-cells, promoting exhaustion. Over time, exhausted T-cells upregulate inhibitory immune checkpoints (PD-1, CTLA-4), while PDAC cells and APCs often overexpress PD-L1. The interaction between PD-L1 and PD-1 transmits suppressive signals, diminishing the release of cytokines such as IFN-γ and cytotoxic functions, thereby facilitating tumor immune escape [19]. This mechanism of T-cell exhaustion induced within the PDAC TIME is illustrated in Figure 1.

Moreover, PDAC is characterized by tumor-infiltrating CD4^+^ and CD8^+^ T-cells; however, as PDAC progresses, the percentage of Tregs elevates within the CD4^+^ T-cell subset, while CD8^+^ T-cells shift in a decreased composition [21,22]. Although T-cells are observed within the TIME of PDAC, it is considered a poorly immune-responsive cancer, as T-cells exhibit a lack of activation or an exhausted phenotype [23].

Furthermore, a recent scRNA-seq study, in combination with multiplex immunohistochemistry imaging, evaluated the T-cell landscape in PDAC, demonstrating that infiltrated CD8^+^ T-cells exhibited a senescent or exhausted phenotype with high expression levels of TIGIT^+^ and CD39^+^ alongside Low/intermediate PD-1 expression [24].

Another scRNA-seq study revealed a reduction in ligand-receptor interactions after chemotherapy, especially between TIGIT on CD8^+^ T-cells and its receptors on cancer cells. This identified TIGIT as a key inhibitory checkpoint molecule, suggesting that chemotherapy may contribute to immunotherapy resistance by impacting the immune interactions within the PDAC TIME [25].

Although T-cell exhaustion is central to PDAC immune evasion, tumor-intrinsic factors such as metabolic reprogramming and extracellular matrix (ECM) remodeling also contribute to immunosuppression and resistance to therapy [19]. Thus, profiling both cancer and immune cell gene expression is essential for understanding PDAC progression.

Therefore, this study aimed to unravel the immune landscape, specifically the T-cell landscape, to identify the novel, exhaustive T-cell markers influenced by tumor-intrinsic gene expression. This study utilized computational methods such as scRNA-seq analysis, differential gene expression, protein–protein interaction analysis, hub-gene identification, and expression validations of the identified hub-genes to gain insights into the molecular and functional aspects of T-cell exhaustion and to identify novel markers of cancer and the T-cells associated with T-cell exhaustion. However, further clinical research is needed to validate the findings of this study.

## 2. Materials and Methods

### 2.1. ScRNA-Seq Dataset Retrieval

The preprocessed (mapped) scRNA dataset of PDAC with the accession ID GSE212966 was retrieved using the database Gene Expression Omnibus (GEO) datasets (https://www.ncbi.nlm.nih.gov/gds accessed on 16 December 2024) of the National Center for Biotechnology Information (NCBI). This public repository archives and distributes functional genomic data [26].

The selected dataset containing all hematoxylin and eosin (H&E)-stained slides from 80 patients were reviewed by a pathologist to confirm PDAC diagnosis, with tumor and adjacent noncancerous regions marked. Duplicate 1.5 mm tissue cores were extracted and assembled into tissue microarrays (TMAs). For scRNA-seq, six PDAC (T1–T6) and six matched adjacent noncancerous resection samples (N1-N6) sequenced using the Illumina NovaSeq 6000 platform were obtained from treatment-naive patients at Peking University First Hospital.

### 2.2. Data Preprocessing and Clustering

The “Seurat version 5.3.0” package within R was utilized for the scRNA-seq analysis of the retrieved dataset, comprising 36,601 genes and 57,167 cells, which were subjected to the quality control measures, including “nFeature_RNA > 200 & <6000”, “nCount_RNA > 1000”, and mitochondrial reads < 10, filtering out the undesired and dead cells.

Moreover, the “LogNormalize” method was used for the normalization of the data, followed by the identification of 2000 highly variable features using the “variance stabilizing transformation (vst)” method. Furthermore, the data was scaled and linear dimensionality reduction was performed through principal component analysis (PCA). Subsequently, the elbow plot was observed, and the first 15 PCs were utilized to identify the cell clusters using “FindNeighbors” and “FindClusters” functions at a resolution of 0.6 and utilizing the Louvain algorithm.

### 2.3. Cell Type Annotation

The identified cell clusters were annotated using the “ScType” database (https://sctype.app/ accessed on 17 December 2024), a computational platform that provides automated cell-type identifications [27]. The “gene_sets_prepare” function was utilized to retrieve immune system-specific gene sets from the ScType database, which prepared both positive and negative marker gene sets. Subsequently, the “sctype_score” function was used to perform annotation based on the expression of prepared marker genes by computing the cell-type-specific scores. These scores were aggregated at the cluster level, allowing for the identification of the most likely cell type associated with each cluster. Additionally, clusters with scores lower than one-fourth of the number of cells in the respective cluster were labeled “Unknown” to ensure annotation confidence.

### 2.4. Molecular Subtypes Classification of PDAC Samples

Subsequently, a subset of cancer cells from the PDAC condition was created to identify the molecular subtypes of all six tumor samples (T1–T6). This was performed to identify the heterogeneity of tumor samples, which can influence the tumor behavior, treatment response, and prognosis.

This was achieved by retrieving the marker genes of five molecular subtypes of PDAC, including classical/pancreatic progenitor, basal-like/squamous/quasi-mesenchymal, immunogenic, aberrantly differentiated endocrine exocrine (ADEX), and activated and normal stroma/stroma-rich PDAC, through the literature. The list of marker genes for all five molecular subtypes is mentioned in Appendix A.

Furthermore, an R version 4.4.2 package, “UCell version 2.10.1” (https://github.com/carmonalab/UCell accessed on 25 January 2025), was utilized to evaluate the gene signatures (retrieved marker genes) in tumor samples (T1–T6). It applies the “Mann–Whitney U statistic” to compute the UCell signature scores, which are robust to data size and heterogeneity. The UCell scoring was applied using the “AddModuleScore_UCell” function, and the enrichment scores for each gene signature were calculated independently for every cell in the cancer cells subset, predicting the molecular subtype of each tumor sample.

### 2.5. Gene Expression Profiling Across Conditions

Moreover, after the identification of cell types, DEGs of cancer cells, CD8^+^ NKT-like cells, memory CD4^+^ T cells, and naive CD4^+^ T cells were identified using the “FindMarkers” function and the “wilcox” method with a threshold of ±0.25.

The differential gene expression analysis of cancer cells was performed in two groups, including cancer cells compared to all other cell types (cancer cells_vs_all-PDAC) within the PDAC condition and cancer cells within the PDAC condition compared to all other cell types in the normal condition (cancer-PDAC_vs_all-normal). Moreover, the CD8^+^ NKT-like cells, memory CD4^+^ T cells, and naive CD4^+^ T cells within the PDAC condition were compared to the same cells in the normal condition (CD8^+^ NKT-like cells-PDAC_vs_CD8^+^ NKT-like cells-normal, memory CD4^+^ T cells-PDAC_vs_memory CD4^+^ T cells-normal, and naive CD4^+^ T cells-PDAC_vs_naive CD4^+^ T cells-normal).

The differential gene expression of the aforementioned combinations of cancer cells was determined to gain insights into the upregulated markers implicated in the heterogeneity and progression of cancer cells by exhausting the T-cells within the TIME. Furthermore, the differentially expressed genes of T-cells (CD8^+^ NKT-like cells, memory CD4^+^ T cells, and naive CD4^+^ T cells) indicated the implication of upregulated markers in the loss of function of T-cells, which might be due to T-cell exhaustion, enabling cancer cells to grow and progress rapidly in the time of PDAC.

Subsequently, common and unique upregulated markers of cancer cells from both groups were distinguished by utilizing Venn diagrams using the web-based tool Venny 2.0 (https://bioinfogp.cnb.csic.es/tools/venny/index2.0.2.html accessed on 25 January 2025).

### 2.6. Pathways Enrichment Analysis, Protein–Protein Interaction (PPI) Analysis, and Hub-Genes Identification

The identified upregulated genes of cancer cells, CD8^+^ NKT-like cells, memory CD4^+^ T cells, and naive CD4^+^ T cells were used for the pathway enrichment analysis using the GeneCodis4 (https://genecodis.genyo.es/ accessed on 3 January 2025), a web-based tool for functional enrichment analysis, allowing researchers to integrate different sources of annotations [28]. The Reactome pathways were selected, and the genes implicated in the top 10 upregulated pathways were used for PPI analysis using STRING version 12.0 (https://string-db.org accessed on 25 January 2025), a database that collects, scores, and integrates all the publicly available sources of PPI information, and to complement these with the computational predictions [29].

Lastly, Cytoscape version 3.10.3 identified the top 10 enriched hub genes from the protein interaction network through the “Degree” method. This open-source software integrates the biomolecular interaction networks with high-throughput expression data [30].

### 2.7. Aberrant Gene Expression Validation and Survival Rates in PDAC Patients

The gene expression validation and survival analysis of the identified hub genes of cancer cells were performed using GEPIA2 (http://gepia2.cancer-pku.cn/#index accessed on 25 January 2025) web server, which is used for large-scale expression profiling and interactive analysis [31]. Moreover, the gene expression validation of CD8^+^ NKT-like cells, memory CD4^+^ T-cells, and naive CD4^+^ T-cells was performed using Tumor Immune Single Cell Hub 2 (TISCH2, http://tisch.comp-genomics.org/ accessed on 25 January 2025), a scRNA-seq data resource from human and mouse tumors, enabling the extensive characterization of gene expression in TIME [32]. Furthermore, this was performed to shortlist the crucial hub genes implicated in the progression of PDAC.

## 3. Results

### 3.1. Data Preprocessing and Cell Type Annotation

The retrieved dataset, consisting of six PDAC samples and six control samples, was merged using the “merge” function in the “Seurat” package, resulting in a combined matrix. This matrix was then converted into a “Seurat Object”, which was further used for quality control measures, retaining 38,589 cells comprising 36,601 genes. Subsequently, normalization, scaling, and highly variable features were identified. Among the 2000 highly variable features, the top 10 were *JCHAIN*, *IGKC*, *SPP1*, *EREG*, *APOC1*, *MZB1*, *IGHG1*, *TPSB2*, *CPA3* and *TPSAB1*. Moreover, linear dimensionality reduction and cell clustering were performed based on gene expression profiles, resulting in 26 cell clusters at a resolution of 0.6. The quality control, highly variable features, PCA plot, elbow plot, and t-SNE of cell clusters are shown in Appendix A.

Furthermore, the cell annotation resulted in 15 cell types, including Basophils, Cancer cells, CD8^+^ NKT-like cells, Endothelial, Macrophages, Memory CD4^+^ T cells, Myeloid Dendritic cells, Naive CD4^+^ T cells, Natural killer cells, Neutrophils, Plasma B cells, Plasmacytoid Dendritic cells, Platelets, Pre-B cells, and Progenitor cells. Interestingly, it was observed that the control condition exhibited trace amounts of cancer cells, indicating the spread of PDAC to control (healthy) regions of the pancreas, leading to metastasis.

Subsequently, it was observed that CD8^+^ NKT-like cells and memory CD4^+^ T cells exhibited a large and dense cluster in the PDAC condition compared to the control condition, suggesting the infiltration of T-cells at the tumor site. However, naive CD4^+^ T cells were observed to have a large and dense cluster of cells in the control condition. Additionally, the cancer cells subset was created to identify the molecular subtypes of PDAC samples. The annotated cell types in the control and PDAC conditions and the cancer cells subset are shown using the t-SNE method in Figure 2a,b.

### 3.2. Molecular Subtypes Classification of PDAC Samples

The R version 4.4.2 package UCell version 2.10.1 (https://github.com/carmonalab/UCell accessed on 25 January 2025) was used to evaluate gene signatures in tumor samples (T1–T6). It calculates signature scores using the “Mann–Whitney U statistic”, offering robustness to data size and heterogeneity. The expression levels of signature genes of respective molecular subtypes in the cancer cell subset are shown in the t-SNE plot in Appendix A. The identification of molecular subtypes indicated heterogeneous profiles of PDAC tumor samples, showing the disparity of tumor samples across multiple molecular subtypes. It was observed that the signature genes of the “activated and normal stroma/stroma-rich” molecular subtype showed distributed cancer cells in trace amounts with moderate expression levels, except for the T5 and T6 samples, which showed low expression levels. However, the T3 and T4 samples showed moderate expression levels with moderately distributed cancer cells. The distribution of cancer cells and the expression levels of signature genes of the “activated and normal stroma/stroma-rich” molecular subtype are shown in Figure 3.

Interestingly, the signature genes of “classical/pancreatic progenitor”, “basal-like/squamous/quasi-mesenchymal”, “immunogenic”, and “ADEX” showed diverse amounts of cancer cells with varying expression levels in all six tumor samples (T1–T6), indicating heterogeneous samples.

The “immunogenic” and “ADEX” signature genes showed moderate cancer cells with low expression levels; however, the signature genes of “classical/pancreatic progenitor” and “basal-like/squamous/quasi-mesenchymal” showed highly distributed dense cancer cells and high expression levels, with cancer cells being slightly more shifted towards the “classical/pancreatic progenitor” molecular subtype in all six tumor samples compared to “basal-like/squamous/quasi-mesenchymal”. The cancer cells’ distribution and expression levels of signature genes of respective molecular subtypes in individual tumor samples are shown in Figure 3.

This suggests that all the tumor samples within the dataset showed a heterogeneous nature, with the “classical/pancreatic progenitor” and “basal-like/squamous/quasi-mesenchymal” molecular subtypes being highly expressed in the majority of the cancer cells. This reveals the aggressive nature of the tumor samples with a diverse genetic makeup.

### 3.3. Aberrant Markers of Cancer Cells, CD8^+^ NKT-like Cells, Memory CD4^+^ T Cells, and Naive CD4^+^ T Cells

The differential gene expression of the “cancer cells_vs_all-PDAC” group resulted in 8509 upregulated and 4642 downregulated genes, totaling 13,151 dysregulated genes. Among these genes, the top 10 upregulated included *AC007529.2*, *LINC02747*, *FGA*, *LGALS7*, *RASSF10*, *FER1L6*, *AC036176.3*, *NPSR1*, *SERPINB3*, and *AADAC* with *p*-values of 3.31 × 10^−47^, 8.77 × 10^−49^, 4.02 × 10^−75^, 3.06 × 10^−49^, 6.12 × 10^−60^, 0, 6.53 × 10^−121^, 0, 0, and 0; meanwhile, the logarithm of fold change (log2FC) was 8.60, 8.12, 7.89, 7.88, 7.82, 7.76, 7.75, 7.72, 7.71, and 7.63, respectively.

Moreover, the “cancer-PDAC_vs_all-normal” group exhibited 9268 upregulated and 4221 downregulated genes, totaling 13,489 dysregulated genes. Among these genes, the top 10 upregulated included *CEACAM5*, *MUC5B*, *SERPINB3*, *KRT6B*, *TFF2*, *AL121761.1*, *TFF1*, *CEACAM6*, *EPS8L3*, and *SPRR3*, with *p*-values of 0, except for *SPRR3*, showing a *p*-value of 1.58 × 10^−108^; meanwhile, these genes showed a log2FC of 14.42, 14.30, 13.77, 13.47, 13.38, 13.11, 13.10, 13.07, 13.01, and 12.60, respectively. The expression levels of both groups of cancer cell genes are shown in Figure 4a.

Furthermore, CD8^+^ NKT-like cells exhibited 2598 upregulated and 3771 downregulated genes, totaling 6369 dysregulated genes. The top 10 upregulated genes included *COL11A1*, *MUC5AC*, *CEACAM5*, *FXYD3*, *MUC5B*, *MUC4*, *KLK6*, *TSPAN1*, *GPX2*, and *CTSE*, with *p*-values of 1.78 × 10^−26^, 5.76 × 10^−30^, 9.61 × 10^−21^, 3.25 × 10^−283^, 1.35 × 10^−16^, 2.17 × 10^−15^, 1.03 × 10^−9^, 1.65 × 10^−8^, 1.90 × 10^−6^, and 2.55 × 10^−28^, while log2FC of 8.81, 8.80, 8.40, 8.23, 8.01, 7.94, 7.35, 7.34, 7.34, and 7.24, respectively.

The memory CD4^+^ T cells showed 2820 upregulated and 3500 downregulated genes, totaling 6320 dysregulated genes. The top 10 upregulated genes included *FXYD3*, *TFF1*, *MUC1*, *OLFM4*, *AGR2*, *LCN2*, *CTSE*, *CEACAM6*, *KRT19*, and *CLDN4*, with *p*-values of 1.53 × 10^−18^, 1.22 × 10^−14^, 1.33 × 10^−11^, 7.02 × 10^−8^, 1.90 × 10^−5^, 6.27 × 10^−5^, 5.09 × 10^−3^, 3.64 × 10^−3^, 7.40 × 10^−13^, and 2.02 × 10^−4^; meanwhile, they had a log2FC of 8.42, 8.17, 7.51, 7.45, 6.97, 6.75, 6.51, 6.47, 6.37, and 5.39, respectively.

The naive CD4^+^ T cells resulted in 4331 upregulated and 3012 downregulated genes, totaling 7343 dysregulated genes. The top 10 upregulated genes included *KRT19*, *TFF1*, *LCN2*, *AGR2*, *TFF2*, *OLFM4*, *CLDN18*, *SPP1*, *C19orf33*, and *CEACAM6*, with *p*-values of 1.14 × 10^−19^, 4.28 × 10^−10^, 7.87 × 10^−11^, 4.89 × 10^−13^, 1.45 × 10^−11^, 3.76 × 10^−7^, 1.45 × 10^−11^, 2.12 × 10^−33^, 7.87 × 10^−11^, and 4.28 × 10^−10^; meanwhile, they had a log2FC of 8.30, 7.95, 7.81, 7.75, 7.72, 7.61, 7.49, 7.49, 7.46, and 7.38, respectively. The expression levels of the top 10 upregulated markers of T-cells are shown in Figure 4b.

It was observed that the “cancer-PDAC_vs_all-normal” group exhibited the greatest and most high intensity dysregulation compared to the “cancer cells_vs_all-PDAC” group, indicating that markers in this group might be implicated in the rapid progression of cancer cells in TIME and may be highly implicated in signaling, leading to T-cell exhaustion. Nevertheless, it was observed that among T-cells, naive CD4^+^ T cells showed the highest number of dysregulated genes, indicating the high influence of cancer cells on naive CD4^+^ T cells.

Subsequently, the upregulated markers of cancer cells showed 7882 common genes among both groups, while 1386 were unique to “cancer-PDAC_vs_all-normal” and 627 were unique to the “cancer cells_vs_all-PDAC” group. Henceforth, all these genes were used separately for further analysis to identify unique and common markers in cancer cells while comparing different conditions. A Venn diagram of the “cancer-PDAC_vs_all-normal” and “cancer cells_vs_all-PDAC” groups is shown in Figure 4c.

### 3.4. Pathways Dysregulation Leading to T-Cell Exhaustion and PDAC Progression

The upregulated markers were used to identify the enriched Reactome pathways. The common markers of cancer cells included Metabolism, Metabolism of lipids, Asparagine N-linked glycosylation, Post-translational protein modification, Membrane Trafficking, RHO GTPase cycle, Vesicle-mediated transport, Regulation of cholesterol biosynthesis by SREBP (SREBF), ER to Golgi Anterograde Transport, and RHOB GTPase cycle as the top 10 enriched pathways.

While the markers unique to the “cancer cells_vs_all-PDAC” group included Gene expression (Transcription), Generic Transcription Pathway, RNA Polymerase II Transcription, Metabolism of RNA, DNA Repair, Antiviral mechanism by IFN-stimulated genes, SMAC (DIABLO) binds to IAPs, SMAC (DIABLO)-mediated dissociation of IAP complexes, Abasic sugar-phosphate removal via the single-nucleotide replacement pathway, and SMAC, XIAP-regulated apoptotic response as the top 10 enriched pathways.

Lastly, the markers unique to the “cancer-PDAC_vs_all-normal” group included Extracellular matrix organization, Collagen formation, Assembly of collagen fibrils and other multimeric structures, Collagen degradation, Collagen biosynthesis and modifying enzymes, Degradation of the extracellular matrix, ECM proteoglycans, Collagen chain trimerization, MET activates PTK2 signaling, and Integrin cell surface interactions as the top 10 enriched pathways. The Reactome pathways of cancer cells markers are shown in Appendix A.

Moreover, the CD8^+^ NKT-like cells exhibited Extracellular matrix organization, Non-integrin membrane-ECM interactions, Assembly of collagen fibrils and other multimeric structures, Collagen formation, Signaling by Receptor Tyrosine Kinases, ECM proteoglycans, MET activates PTK2 signaling, Immune System, MET promotes cell motility, and Syndecan interactions as the top 10 upregulated pathways.

Furthermore, the top 10 upregulated pathways of memory CD4^+^ T cells included Signaling by Receptor Tyrosine Kinases, Neutrophil degranulation, Immune System, Syndecan interactions, Assembly of collagen fibrils and other multimeric structures, Potential therapeutics for SARS, Collagen degradation, Extracellular matrix organization, Disease, and TP53 Regulates Transcription of Genes Involved in G2 Cell Cycle Arrest.

Lastly, the top 10 enriched pathways of naive CD4^+^ T cells were Gene expression (Transcription), Metabolism of RNA, Post-translational protein modification, Immune System, Processing of Capped Intron-Containing Pre-mRNA, RNA Polymerase II Transcription, Cytokine Signaling in Immune system, Vesicle-mediated transport, Membrane Trafficking, and mRNA Splicing. The upregulated reactome pathways of T-cell markers are shown in Appendix A.

Notably, all the cancer cells’ pathways were uniquely implicated in cancer cells’ ability to evade the immune system and promote their growth and progression in the TIME by exhausting the T-cells. Moreover, it was observed that the “Immune system” pathway was shared among all T-cells, while the “Extracellular matrix organization”, “Assembly of collagen fibrils and other multimeric structures”, “Signaling by Receptor Tyrosine Kinases”, and “Syndecan interactions” pathways were found to be common among CD8^+^ NKT-like cells and memory CD4^+^ T cells.

This suggested that cancer cells were sending similar signals to upregulate similar pathways in CD8^+^ NKT-like cells and memory CD4^+^ T cells. However, naive CD4^+^ T-cells did not exhibit common pathways with other T-cells except for the “Immune system”, suggesting the exhaustion of all three T-cells in this study, eventually leading to the suppression of the immune system. This also indicates the heterogeneity with which cancer cells dysregulate the immune cells (T-cells) through specific signals to specific T-cells, which creates complex interactions within the TIME of PDAC.

### 3.5. Key Candidate Proteins Associated with T-Cell Exhaustion Within PDAC TIME

The genes from the top 10 significant pathways of each cell type were subjected to PPI analysis, resulting in different networks of proteins interacting with each other. The upregulated 1960 common pathway genes of cancer cells resulted in a network of 1949 proteins interacting. Moreover, 124 upregulated genes of the “cancer cells_vs_all-PDAC” group exhibited a network of 91 proteins, while 69 upregulated genes of the “cancer-PDAC_vs_all-normal” group resulted in a network of 69 proteins interacting with each other. The PPI networks of cancer cell genes are shown in Appendix A.

Furthermore, 520 upregulated genes of CD8^+^ NKT-like cells exhibited a network of 513 proteins. In comparison, 657 enriched genes of memory CD4^+^ T cells showed a network of 639 proteins, and 1454 upregulated genes of naive CD4+ T cells showed a network of 1445 proteins. The protein networks of CD8^+^ NKT-like cells, memory CD4^+^ T cells, and naive CD4^+^ T cells are shown in Appendix A.

Subsequently, these protein networks of each cell type were used to identify the most interacting genes (hub genes) using the “Degree” method. The protein network of common cancer cell genes showed the top 10 hub genes, including GAPDH, AKT1, EGFR, CS, RHOA, TPI1, SDHA, TFRC, FASN, and HIF1A; the network showed 364, 284, 242, 212, 207, 200, 189, 189, 186, and 179 interactions, respectively.

The protein network of unique cancer cell genes in the “cancer cells_vs_all-PDAC” group exhibited top 10 hub genes, including *H4C6* (*HIST1H4A* or *HIST1H4C*), *MYC*, *H3C12* (*HIST1H3A* or *HIST1H3D*), *DDX21*, *USP7*, *RFC4*, *APEX1*, *CDK9*, *H2BC9* (*HIST1H2BH*), and *NOP2*. Among these proteins, *RFC4*, *APEX1*, *CDK9*, *HIST1H2BH*, and *NOP2* showed 12 interactions, while *DDX21* and *USP7* showed 16 interactions; lastly, *H4C6*, *MYC*, and *H3C12* exhibited 29, 28, and 18 interactions, respectively.

The protein network of unique cancer cell genes in the “cancer-PDAC_vs_all-normal” group showed top 10 hub genes, including *FN1*, *COL1A1*, *COL1A2*, *COL3A1*, *COL5A2*, *COL6A1*, *COL5A1*, *BGN*, *COL6A2*, and *FBN1*, showing 62, 59, 56, 55, 51, 50, 50, 48, 46, and 45 interactions within the protein network, respectively. The hub genes of cancer cells are shown in Figure 5a–c. The common and unique hub genes of cancer cells from both groups are mentioned in Appendix A.

Moreover, the protein network of CD8^+^ NKT-like cells exhibited *TP53*, *FN1*, *MMP9*, *CD4*, *IFNG*, *NFKB1*, *HIF1A*, *HSP90AA1*, *ITGB1*, and *HSP90AB1* as the top 10 hub genes, showing 176, 170, 149, 147, 147, 142, 129, 126, 119, and 116 interactions, respectively. Furthermore, *AKT1*, *TP53*, *ACTB*, *CD4*, *JUN*, *FN1*, *MMP9*, *HSP90AA1*, *MAPK3*, and *HSP90AB1* were the top 10 hub genes in memory CD4^+^ T cells, exhibiting 200, 192, 180, 148, 146, 142, 126, 123, 121, and 113 interactions. Lastly, the protein network of naive CD4^+^ T cells showed *TP53*, *UBC*, *UBB*, *HSP90AA1*, *JUN*, *CTNNB1*, *NFKB1*, *HSP90AB1*, *HSPA8*, and *H3-3B* (*H3F3B*) top 10 hub genes, showing 452, 298, 279, 270, 260, 260, 251, 238, 229, and 212 interactions, respectively. The hub genes of T-cells are shown in Figure 5d–f. The hub genes of all three T-cells are mentioned in Appendix A.

Notably, the sub-groups of cancer cells (common genes, cancer cells_vs_all-PDAC, and cancer-PDAC_vs_all-normal) indicated the unique functional implication of respective hub-genes within their specific group. This suggests the complexity and heterogeneity of cancers within the TIME, as multiple genes are implicated in the progression of PDAC cancer via several factors, including T-cell exhaustion.

Subsequently, it was observed that *TP53*, *HSP90AA1*, and *HSP90AB1* were common among all three T-cells; *FN1*, *MMP9*, and *CD4* were common in CD8^+^ NKT-like cells and memory CD4^+^ T cells; *NFKB1* was common in CD8^+^ NKT-like cells and naive CD4^+^ T cells; and *JUN* was common in memory CD4^+^ T cells and naive CD4^+^ T cells. This indicated that multiple hub-genes of T-cells were common, suggesting that cancer cell markers might be implicated in upregulating common markers of different T-cells through various pathways essential for these T-cells. The expression of the hub genes of cancer cells and T-cells is shown in a heatmap in Figure 6.

Additionally, the unique markers of T-cells indicated that cancer cells influence those markers of T-cells unique to respective T-cell types to evade the immune response through T-cell exhaustion. Appendix A show the expression of cancer cells and T-cell hub genes across different cell types exhibited in dot plots, violin plots, and box plots.

Overall, the dysregulation at the gene and corresponding protein expression levels can impair crucial pathways, as *TP53* governs DNA repair, apoptosis, and cell cycle arrest, with dysregulation compromising genomic integrity. *FN1* and *MMP9* mediate extracellular matrix interactions and degradation, promoting tissue remodeling, invasion, and inflammation when overexpressed. *CD4* functions as a TCR co-receptor; its alteration can impair immune activation. *IFNG* drives Th1 responses and macrophage activation, with imbalances contributing to chronic inflammation. *NFKB1* regulates inflammatory and survival pathways, and its overactivation may lead to autoimmunity. *HIF1A* modulates hypoxia response and metabolism, aiding tumor adaptation under low oxygen.

*HSP90AA1*, *HSP90AB1*, and *HSPA8* encode chaperones crucial for stabilizing client proteins, whose dysregulation supports oncogenic signaling and impairs stress responses. *ITGB1* controls adhesion and migration, with aberrant expression affecting immune trafficking and metastasis. *AKT1*, *MAPK3*, and *JUN* are central to survival, proliferation, and inflammation via the *PI3K/AKT*, *MAPK*, and *AP-1* pathways. *ACTB* ensures cytoskeletal integrity, while *UBC*/*UBB* regulates proteostasis through ubiquitin-mediated degradation. *CTNNB1* integrates Wnt signaling and cell adhesion and *H3F3B* influences transcription via chromatin remodeling. Conclusively, the dysregulation of the expression level of these proteins may disrupt immune signaling, stress adaptation, and cellular homeostasis, contributing to T-cell exhaustion through sustained activation, metabolic stress, and the loss of effector function.

### 3.6. Expression Profiling and Survival Correlation of Key Hub-Genes

GEPIA2 was used for the expression validation and survival analysis of cancer cell hub genes, as it contains TCGA bulk-RNA datasets. At the same time, TISCH2 was utilized for the expression validation of T-cells and cancer cell hub genes, as TISCH2 contains the scRNA-seq datasets. Moreover, GEPIA2 showed all cancer cell hub genes to be highly upregulated in the tumor compared to the control samples. The expression values of all the hub genes of cancer cells in the tumor and control samples are mentioned in Appendix A.

Furthermore, the survival analysis of the cancer cells hub-genes indicated that GAPDH, AKT1, CS, RHOA, TPI1, SDHA, FASN, HIF1A, FN1, COL1A1, COL1A2, COL3A1, COL5A2, COL5A1, BGN, COL6A2, FBN1, USP7, CDK9, H2BC9, and NOP2 exhibited insignificant *p*-values. In contrast, EGFR, TFRC, COL6A1, MYC, DDX21, RFC4, and APEX1 exhibited significant *p*-values, including 0.03, 0.04, 0.04, 0.01, 0.01, 0.04, and 0.04, respectively. Lastly, GEPIA2 showed no results for the overall survival analysis of H4C6 and H3C12.

Subsequently, the significant hub-genes, including *EGFR*, showed that high expression was associated with ~18% of patients’ survival for more than 60 months, while low expression was associated with ~20% of patients’ survival for over 90 months. Moreover, the high expression of *TFRC* was associated with the survival of ~18% of patients for more than 70 months, while low expression was associated with the survival of ~18% for more than 90 months. Furthermore, the high expression of *COL6A1* indicated that none of the patients survived more than 70 months, while ~37% of the low-expression patients survived for more than 90 months.

The high expression of the MYC showed that ~11% of patients survived for more than 70 months, while ~29% of low-expression patients survived for more than 90 months. The high expression of the *DDX21* showed that none of the patients survived more than 75 months, while ~38% of the low-expression patients survived more than 90 months. The high expression of *RFC4* showed ~13% of patients survived for ~80 months, while ~32% survived for more than 90 months. Lastly, the high expression of *APEX1* showed that none of the patients survived for more than 75 months, while ~35% survived for more than 90 months. The overall survival plots of all the cancer cells’ hub genes are shown in Figure 7.

Additionally, the TISCH2 indicated that all the hub-genes of cancer cells were upregulated in malignant cells, especially *GAPDH*, *RHOA*, *TPI1*, *H4C6*, *DDX21*, and *APEX1*, which showed considerably high expression levels in malignant cells. Moreover, the hub-genes of T-cells indicated the upregulation of all hub-genes, especially *ACTB*, *H3F3B*, *HSP90AA1*, *HSP90AB1*, *HSPA8*, *UBB*, and *UBC*, which exhibited high levels of expression in CD8Tex cells. The feature plots of cancer cells and T-cells hub genes from TISCH2 are shown in Appendix A. The violin plots of cancer cells and T-cells hub-genes are shown in Appendix A.

This indicated that the hub genes of cancer cells and T-cells showed high expression in datasets from GEPIA2 and TISCH2. However, the insignificant *p*-values in the overall survival rate of the cancer cell hub genes further need corroborations as these genes are well-reported to be expressed in PDAC patients, leading to poor prognosis.

## 4. Discussion

There is a high rate of recurrence and metastasis in PDAC patients after surgery, lacking efficient chemotherapy, radiotherapy, and immunotherapy [33]. T-cell exhaustion, which is induced by the complex immunosuppressive TIME, may contribute to diminished immunotherapy responses in PDAC patients [34]. Therefore, this study aimed to identify the upregulated markers of cancer cells and T-cells implicated in the progression of PDAC through T-cell exhaustion by utilizing scRNA-seq analysis.

Subsequently, this study identified heterogeneous tumor samples, mainly classified into “classical/pancreatic progenitor” and “basal-like/squamous/quasi-mesenchymal” molecular subtypes, which provided significant insights into the underlying biological diversity of PDAC. Moreover, the present study showed the enriched pathways of cancer cells implicated in crucial biological processes for maintaining cellular homeostasis, including lipid and protein metabolism, post-translational modifications, and membrane trafficking [35,36,37]. Moreover, pathways such as regulation of gene expression, RNA metabolism, and DNA repair are essential for efficient cell function and survival [38,39,40]. This suggests that the cancer cells’ pathways collectively play a vital role in maintaining cellular integrity, communication, and responses to external signals, leading to cancer cells’ progression by evading the immune system.

Furthermore, the enriched pathways of CD8^+^ NKT-like cells, memory CD4^+^ T-cells, and naive CD4^+^ T-cells were implicated in gene regulation, cellular communication, and immune responses. The paths, including receptor tyrosine kinases and MET signaling pathways, impact immune responses and cell movement by regulating cell motility, growth, and survival [41,42]. Additionally, immune-related pathways such as neutrophil degranulation and cytokine signaling were highly expressed in T-cells [43,44]. This indicated that the high expression of immune-related pathways might lead to T-cell exhaustion, rendering them non-functional against cancer cells.

Notably, all three sub-groups (common, cancer cells vs. all-PDAC, and cancer-PDAC vs. all-normal) of cancer cells exhibiting hub-genes indicated the heterogeneity and complexity of cancer cells within the TIME. Moreover, all the hub-genes, including *FN1*, *COL1A1*, *COL1A2*, *COL3A1*, *COL5A2*, *COL6A1*, *COL5A1*, *BGN*, *COL6A2*, and *FBN1*, of the “cancer-PDAC_vs_all-normal” group are reported to be implicated in PDAC patients, leading to poor prognosis [45,46,47,48,49,50,51].

Similarly, common hub-genes such as *GAPDH*, *AKT1*, *EGFR*, *RHOA*, *TPI1*, *SDHA*, *TFRC*, *FASN*, and *HIF1A* are reported to be implicated in the oncogenic pathways leading to cancer cells’ proliferation and progression, playing a crucial role in PDAC [52,53,54,55,56,57,58,59,60], while *CS* is not explicitly reported in PDAC, resulting in a novel PDAC cancer cell marker.

Furthermore, the *MYC*, *DDX21*, *USP7*, *RFC4*, *APEX1*, *CDK9*, and *NOP2* of the “cancer cells_vs_all-PDAC” group play a critical role in the metabolic reprogramming of the PDAC, contributing to the poor prognosis of PDAC [61,62,63,64,65,66,67]. However, *H4C6*, *H3C12*, and *H2BC9* are not reported in cancer cells of PDAC patients, and these novel markers might be implicated in cell proliferation and survival of cancer cells by sending signals, leading to the exhaustion of T-cells.

Additionally, among the common hub-genes of all three or any two T-cell subtypes (CD8^+^ NKT-like cells, memory CD4^+^ T-cells, and naive CD4^+^ T-cells), *CD4* is reported to be activated in T-effs and strongly promotes EMT-associated alterations in H6c7 cells and increased invasive behavior [68]. NFKB1 within the NFKB signaling pathway influences cytotoxic T-cells in the blood of patients with PDAC [69]. *HSP90AA1* has been reported to be expressed in a new type of T-cell (HSP T or Thsp) within the PDAC environment [70]. In contrast, *JUN* is reported to be implicated in the T-cell exhaustion mechanism within the PDAC environment [19].

However, *TP53*, *MMP9*, *FN1*, and *HSP90AB1* were observed to be novel markers, as these markers are not explicitly reported in CD8^+^ NKT-like cells, memory CD4^+^ T-cells, naive CD4^+^ T-cells, or any related T-cell within PDAC patients. This indicates the involvement of these novel markers in T-cell exhaustion and aberrant functional activity within PDAC TIME, eventually leading to PDAC cell proliferation and survival.

Moreover, the unique hub-genes of CD8^+^ NKT-like cells showed that *IFNG* and *HIF1A* are implicated in T-cells [71,72], while *ITGB1* turned out to be a novel marker of CD8^+^ NKT-like cells. Furthermore, the unique hub-genes of memory CD4^+^ T-cells, such as AKT1, ACTB, and MAPK3, were observed to be novel markers as these are not explicitly reported in the T-cells of PDAC patients. Lastly, unique hub-genes of naive CD4^+^ T-cells, such as HSPA8, are reported to be expressed in CD8^+^ T-cells in prostate cancer [73], while *UBC*, *UBB*, *CTNNB1*, and *H3-3B* are not reported, rendering them novel markers for naive CD4^+^ T-cells. This suggests that the aforementioned novel markers are potentially implicated in the exhaustion of T-cells within TIME, eventually leading to cancer cells’ growth, proliferation, and survival by evading the immune responses in PDAC patients.

In this study, CD8^+^ NKT-like, memory CD4^+^ T-cells, and naive CD4^+^ T-cells showed the enrichment of pathways such as cytokine signaling, receptor tyrosine kinase signaling, and neutrophil degranulation; these are biological processes that, when chronically activated, can drive T-cell exhaustion. The upregulation of immune regulatory genes like *JUN*, *NFKB1*, and *HSP90AA1* in these subsets indicates sustained activation and metabolic stress, impairing cytotoxic function in CD8^+^ T-cells and helper capacity in CD4^+^ T-cells. This dysfunctional phenotype likely weakens anti-tumor immunity, facilitating immune escape and PDAC progression.

Subsequently, the expression and survival analysis of the cancer cells and T-cell hub-genes showed that the majority of the cancer cells’ overall survival was insignificant (*p*-value > 0.05); however, the studies above support that the aberrant expression of these markers is implicated in the poor prognosis associated with PDAC. Further corroboration is needed, especially for the novel markers of cancer cells and T-cells within PDAC patients for diagnostic, prognostic, and therapeutic interventions.

Recent scRNA-seq studies on T-cells within PDAC patients have reported T-cell marker genes in CD8^+^ Tcm, CD4^+^ Tem, and γδT cells, and noted that the *CCL5/SDC1* receptor–ligand interactions in tumor-infiltrating T-cells could promote tumor cell migration [74,75]. However, this study reveals the novel biomarkers of cancer cells, CD8^+^ NKT-like cells, memory CD4^+^ T-cells, and naive CD4^+^ T-cells implicated in T-cell exhaustion, inducing the growth, proliferation, and progression of cancer cells.

In summary, the aforementioned novel biomarkers of heterogeneous cancer cells and T-cells might be involved in the complex TIME, leading to the poor prognosis of PDAC. These findings underscore the complexity and heterogeneity of the TIME and its influence on immune evasion and cancer cell survival. While existing studies support the role of some identified markers in poor prognosis, further experimental validation before clinical applications is required for PDAC management.

## 5. Conclusions

This study revealed novel biomarkers of heterogeneous cancer cells (*CS*, *H4C6*, *H3C12*, and *H2BC9*) and T-cells (*TP53*, *FN1*, *MMP9*, *ITGB1*, *HSP90AB1*, *AKT1*, *ACTB*, *MAPK3*, *UBC*, *UBB*, *CTNNB1*, and *H3-3B*), including CD8^+^ NKT-like cells, memory CD4^+^ T-cells, and naive CD4^+^ T-cells, which might be the key candidate biomarkers implicated in cancer cell progression through T-cell exhaustion. Further experimental validations before clinical applications are needed to evaluate these novel biomarkers as potential therapeutic options in PDAC patients.

## Figures and Tables

**Figure 1 biology-14-01015-f001:**
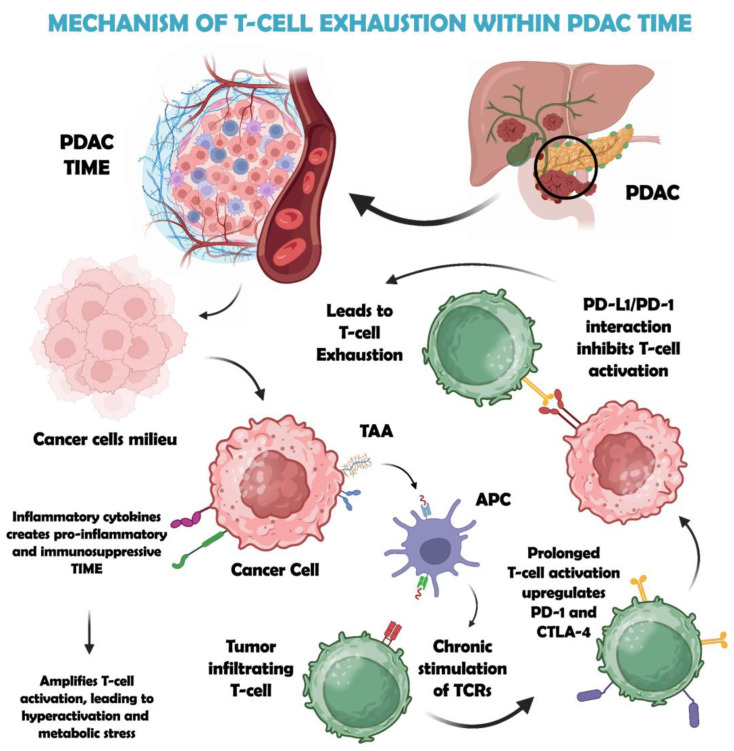
The mechanism of T-cell exhaustion induced within the PDAC tumor immune microenvironment (TIME). It shows that in PDAC, cancer cells release tumor-associated antigens (TAAs) into TIME, where they are taken up by antigen-presenting cells (APCs) such as dendritic cells and macrophages. These APCs process the TAAs and present them on major histocompatibility complex class II (MHC-II) molecules to CD4^+^ helper T-cells. This interaction leads to chronic T-cell receptor (TCR) stimulation. This triggers T-cell hyperactivation, amplified by PDAC-secreted cytokines like IL-6 and TGF-β, leading to metabolic stress. Prolonged activation induces the upregulation of inhibitory receptors (PD-1, CTLA-4) on T-cells, while PDAC cells overexpress PD-L1. The PD-L1/PD-1 interaction suppresses T-cell effector functions, reducing cytokine production (e.g., IFN-γ) and cytotoxicity, enabling tumor immune evasion.

**Figure 2 biology-14-01015-f002:**
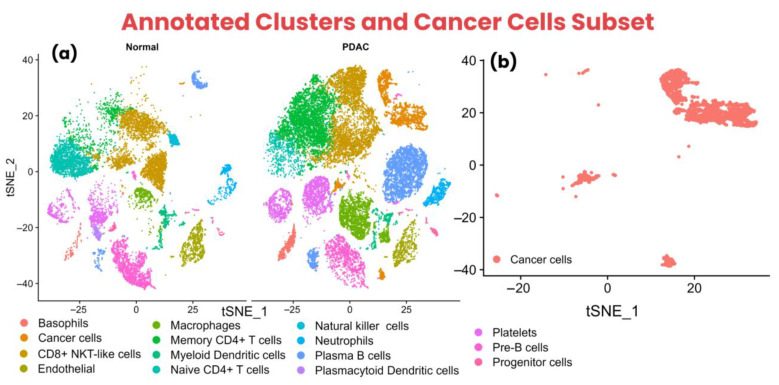
Cell type annotation clusters are shown using a t-SNE plot. (**a**) The annotated clusters within normal and PDAC conditions show different cell populations in both conditions, indicating the complex influence of cancer cells in the PDAC condition compared to the normal condition. (**b**) The cancer cells subset used to identify molecular subtypes of PDAC tumor samples (T1–T6).

**Figure 3 biology-14-01015-f003:**
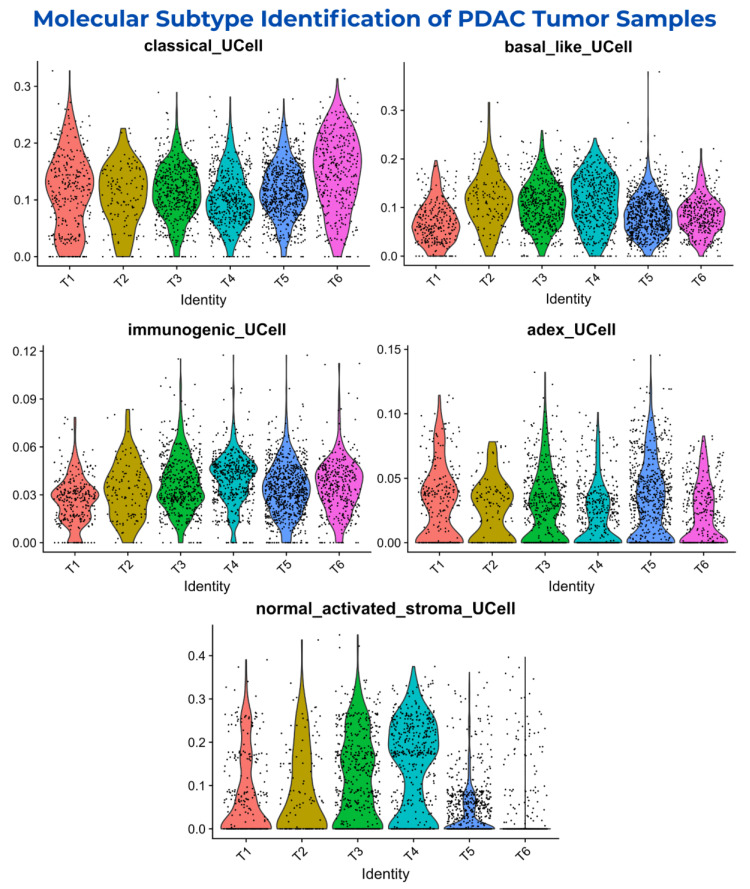
The identification of “classical/pancreatic progenitor”, “basal-like/squamous/ quasi-mesenchymal”, “immunogenic”, “ADEX”, and “activated and normal stroma/stroma-rich” molecular subtypes of PDAC tumor samples (T1–T6). The signature genes of “classical/pancreatic progenitor” and “basal-like/squamous/quasi-mesenchymal” showed expression levels ranging from 0 to 0.3, while “immunogenic” and “ADEX” ranged from 0 to 0.12 and from 0 to 0.15. Lastly, the signature genes of “activated and normal stroma/stroma-rich” showed expression levels ranging from 0 to 0.4 with trace amounts of cancer cells. This indicated the heterogeneity of all six tumor samples in the dataset.

**Figure 4 biology-14-01015-f004:**
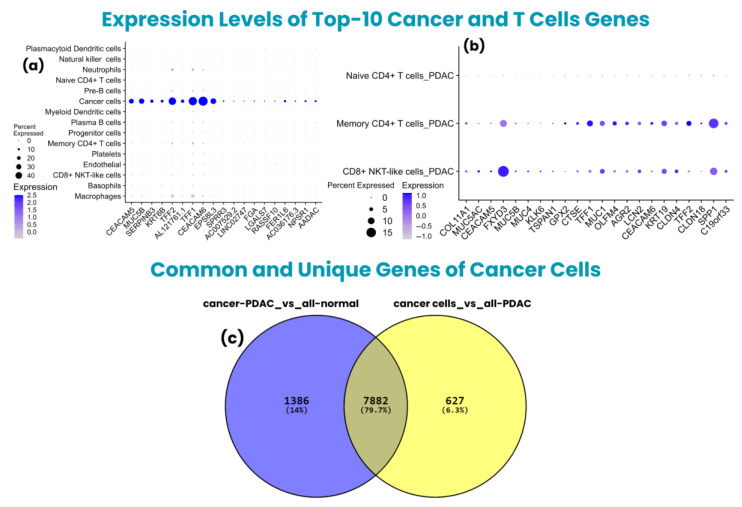
Expression plots of top 10 cancer and T-cell genes, along with common and unique genes of cancer cells groups. (**a**) Dot plot of the top 10 cancer cell genes showing high expression explicitly in cancer cells and very low expression levels in other cell types, indicating crucial cell-type-specific expression, which might be implicated in tumor progression, (**b**) dot plot of top 10 T-cell subset genes (naive CD4^+^ T-cells, memory CD4^+^ T-cells, and CD8^+^ NKT-like cells) showing high expression in memory CD4^+^ T-cells and CD8^+^ NKT-like cells, while a low level of upregulation in naive CD4^+^ T-cells, as few tumor samples showed upregulation; therefore exhibiting overall low level of upregulation, (**c**) Venn diagram of cancer cells genes of “cancer-PDAC_vs_all-normal” and “cancer cells_vs_all-PDAC” groups, indicating the common and unique genes among both groups.

**Figure 5 biology-14-01015-f005:**
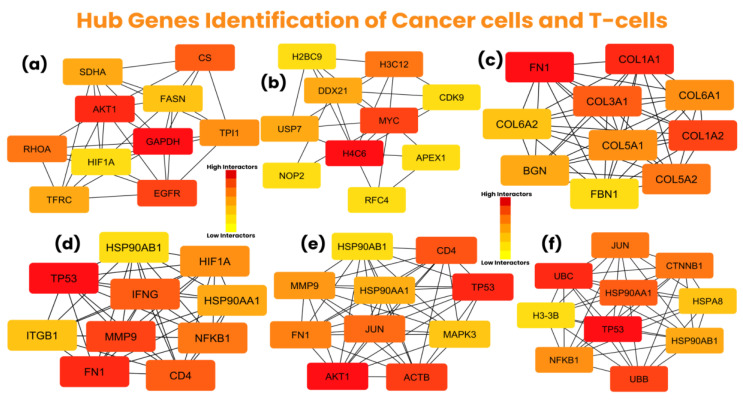
Hub gene identification of cancer cells and T-cells using Cytoscape version 3.10.3. (**a**) The top 10 hub genes of the “common cancer cells” group, with *GAPDH* exhibiting the most interactions, (**b**) the top 10 hub genes of the “cancer cells_vs_all-PDAC” group, with *H4C6* showing the highest number of interactions, (**c**) the top 10 hub genes of the “cancer-PDAC_vs_all-normal” group, with *FN1* showing the most interactions, (**d**) the top 10 hub genes of CD8^+^ NKT-like cells, with *TP53* as the most interacting hub gene, (**e**) the top 10 hub genes of memory CD4^+^ T cells, with *AKT1* exhibiting most interactions, (**f**) the top 10 hub genes of naive CD4^+^ T cells, with *TP53* as the most interacting hub gene.

**Figure 6 biology-14-01015-f006:**
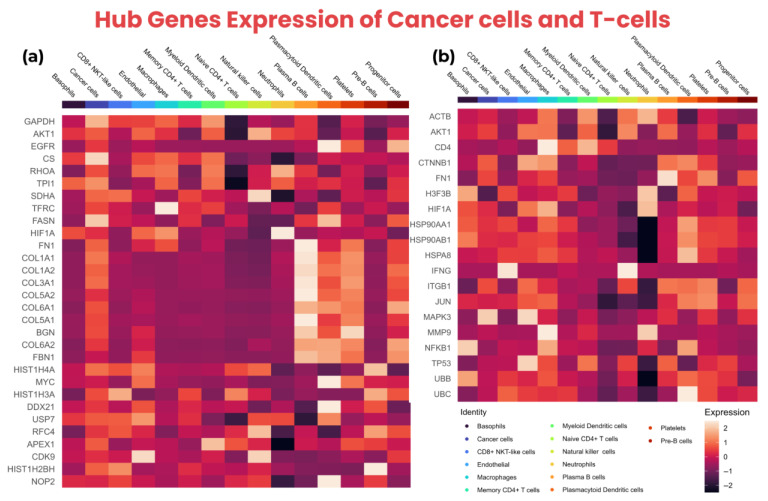
The expression of hub genes in cancer cells and T-cells is shown using a heatmap. (**a**) The heatmap of the top 10 hub genes from all three groups (common markers of cancer cells, cancer cells_vs_all-PDAC, and cancer-PDAC_vs_all-normal) showing high expression across multiple cell types, (**b**) the heatmap of the top 10 hub genes of T-cells showing expression in various cell types including CD8^+^ NKT-like cells, memory CD4^+^ T cells, and naive CD4^+^ T cells.

**Figure 7 biology-14-01015-f007:**
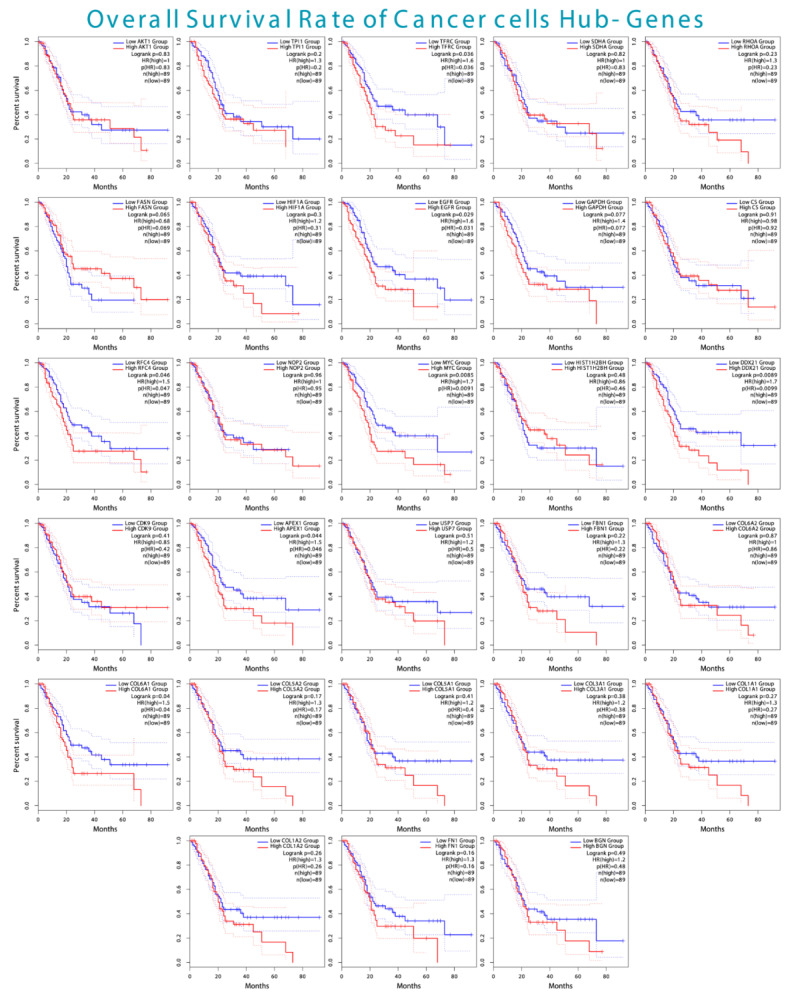
The overall survival rate of all significant and insignificant cancer cell hub-genes, indicating the high and low expressions, leading to the overall survival rate in months.

## Data Availability

All data utilized in this study are publicly accessible. The single-cell RNA-seq dataset analyzed is available from the Gene Expression Omnibus (GEO) under accession ID GSE212966. The R scripts used for dataset preprocessing and analysis are available via Seurat’s official GitHub repository (https://github.com/satijalab/seurat accessed on 25 January 2025). Source code for cell type annotation of cell clusters is accessible through the GitHub repository of the ScType database (https://github.com/IanevskiAleksandr/sc-type accessed on 25 January 2025). Code used for molecular subtype classification of PDAC samples is available from the GitHub repository of UCell: Robust and Scalable Single-Cell Gene Signature Scoring (https://github.com/carmonalab/UCell accessed on 25 January 2025).

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
