# Peer review of "Novel Insights into T-Cell Exhaustion and Cancer Biomarkers in PDAC Using ScRNA-Seq"

_biology, 2025, doi:10.3390/biology14081015_

Round 1

Reviewer 1 Report

Comments and Suggestions for Authors

The study by Saleem and colleagues presents RNA-seq analysis of PDAC samples from a publicly available database. While the technique is innovative, the study lacks novelty. As acknowledged by the authors, previous studies have already conducted scRNA-seq on PDAC samples and reported increased expression of T cell exhaustion markers. A major concern is the absence of protein-level validation. Furthermore, the manuscript is written in a highly technical style, which hinders the biological interpretation of the results.

Abstract

  • It’s not specified where the cells come from, if they were isolated from tumor sample, from human or mouse. More details are needed.

Introduction

  • In lines 44-45, where do these data come from? Is it global statistics or from the US? If so, please add this information, as it can be misleading or replaced with global data.
  • Lines 46-49: There is a repetition of “in the United States”, please delete one.
  • You could add a sentence explaining why PDAC is so deadly, informing that it is due to the late diagnosis, for example.
  • The information about immunotherapy, beginning on lines 61-6,5 should be added to the following paragraph (lines 66-71).
  • The whole Figure 1 mechanism should be better described in the introduction section.
  • In the last paragraph, another study that performed sc-RNAseq in PDAC tissue is mentioned. What are the differences between this and the present study? What is the relevance of the present study?
  • In general, it seems that the main topic of the manuscript is T cell exhaustion, but the results are very broad in identifying relevant genes expressed in PDAC, not only related to immune cells. If you’re describing relevant genes expressed by PDAC cells, you could include some information in the introduction.

Figure 1:

  • You should spell out TIME in the first line, not in the second. Also, delineate the title of the figure with a dot, and then start describing in another paragraph.
  • Spell out TCR, and explain the binding to MHC-II molecules by the APCs.

Methods

  • What are the control samples? Are they healthy tissue from the pancreas?
  • Is there any detail about the tumor tissue or the patients?

Results

  • Please revise figure 2, as some cell titles are on top of the other, which unable proper reading.
  • The results in lines 237-247 refer to which figure? There is no indication.
  • What is ADEX in line 245?
  • How was the cancer cell classification done? They are exposed in Figure 3, but there is no explanation about them.
  • Sentence in lines 259-261 should be the first in section 3.2.
  • The authors should consider presenting the data from section 3.3 in a figure in the main manuscript, not in the supplementary.
  • The authors should relate genes to the proteins and their functions, so their findings would have a more relevant meaning.
  • They should add a color scale to explain what the color difference represents in Figure 4.
  • If most of the data is in the supplementary, it’s hard to follow what is described in the Results section.

Author Response

Thank you very much for taking the time to review this manuscript. I've included the detailed responses below and the revisions/corrections highlighted/in-track changes in the resubmitted files for you to review.

Comments and Suggestions for Authors

The study by Saleem and colleagues presents RNA-seq analysis of PDAC samples from a publicly available database. While the technique is innovative, the study lacks novelty. As acknowledged by the authors, previous studies have already conducted scRNA-seq on PDAC samples and reported increased expression of T cell exhaustion markers. A major concern is the absence of protein-level validation. Furthermore, the manuscript is written in a highly technical style, which hinders the biological interpretation of the results.

Abstract

  • It’s not specified where the cells come from, if they were isolated from tumor samples, from human or mice. More details are needed.

Answer: Agree. This critical point is now addressed in lines 20-23, as the publicly available human PDAC dataset was used, with cells isolated from primary tumors and adjacent normal tissues.

Introduction

  • In lines 44-45, where do these data come from? Is it global statistics or from the US? If so, please add this information, as it can be misleading or replaced with global data.

Answer: Thank you for pointing this out. These statistics are from the US and are now addressed in line 50.

  • Lines 46-49: There is a repetition of “in the United States”, please delete one.

Answer: Thank you for pointing this out. Addressed and deleted the repetition.

  • You could add a sentence explaining why PDAC is so deadly, informing that it is due to the late diagnosis, for example.

Answer: Agree. A sentence explaining why PDAC is so deadly was added in lines 43-45.

  • The information about immunotherapy, beginning on lines 61-65, should be added to the following paragraph (lines 66-71).

Answer: Agree. Information about immunotherapy was added to the paragraph starting at line 67.

  • The whole Figure 1 mechanism should be better described in the introduction section.

Answer: Thank you for pointing this out. Figure 1: The mechanism is explained in the introduction section in lines 79-91.

  • In the last paragraph, another study that performed sc-RNAseq in PDAC tissue is mentioned. What are the differences between this and the present study? What is the relevance of the present study?

Answer: Thank you for pointing this out. The present study focuses on identifying novel markers of T-cells and cancer cells involved in T-cell exhaustion, leading to cancer cell progression in PDAC using scRNA-seq, specifically emphasizing T-cell subtypes and immune evasion mechanisms. In contrast, the other study broadly characterizes the PDAC tumor microenvironment and examines how chemotherapy alters immune and stromal cell populations. While the present study highlights novel biomarkers and exhaustion pathways, the other research emphasizes treatment-induced remodelling and suggests TIGIT as a promising immunotherapy target.

Relevance of the present study

            The present study profiled exhaustion-associated pathways and hub genes in T-cells, revealing how PDAC induces immune dysfunction. It also identified niche-specific cancer cell markers that might influence T-cells towards exhaustion. It helps fill a gap in the current PDAC immunotherapy literature by exploring under-characterized T-cell subpopulations beyond CD8⁺ exhaustion, including NKT-like and memory CD4⁺ cells. Identifying these markers provides potential therapeutic targets and biomarkers for future development of precision immunotherapies, especially those targeting T-cell exhaustion in TIME. Unlike broader studies, this study highlights the heterogeneity in immune exhaustion signatures, which can be leveraged to tailor immunotherapeutic approaches for PDAC patients effectively.

  • In general, it seems that the main topic of the manuscript is T cell exhaustion, but the results are very broad in identifying relevant genes expressed in PDAC, not only related to immune cells. If you’re describing relevant genes expressed by PDAC cells, you could include some information in the introduction.

Answer: Agree. Added literature about PDAC intrinsic gene expression relevant to T-cell exhaustion in lines 107-111.

Figure 1:

  • You should spell out TIME in the first line, not in the second. Also, delineate the title of the figure with a dot, and then start describing in another paragraph.

Answer: Thank you for pointing this out. Spelled TIME in the first line, delineated the title, and described Figure 1 in another paragraph.

  • Spell out TCR, and explain the binding to MHC-II molecules by the APCs.

Answer: Agree. Spelled TCR and explained the binding of MHC-II molecules by the APCs.

Methods

  • What are the control samples? Are they healthy tissue from the pancreas?

Answer: Thank you for pointing this out. Control samples refer to the healthy tissues from the pancreas.

  • Is there any detail about the tumor tissue or the patients?

Answer: Thank you for pointing this out. The details about the patients and tumor tissues are updated in lines 139-145.

Results

  • Please revise figure 2, as some cell titles are on top of the other, which unable proper reading.

Answer: Agree. Figure 2 is revised, and all the cell types are mentioned in the color legend

  • The results in lines 237-247 refer to which figure? There is no indication.

Answer: Thank you for pointing this out. These results refer to Figure 3 and are updated in lines 278-279.

  • What is ADEX in line 245?

Answer: ADEX stands for “aberrantly differentiated endocrine exocrine”. It is a molecular subtype of PDAC and is first abbreviated in section 2.4.

  • How was the cancer cell classification done? They are exposed in Figure 3, but there is no explanation about them.

Answer: Agree. Section 2.4 already explains the cancer cell classification. However, brief details are added in the results section 3.2, lines 267-269.

  • Sentence in lines 259-261 should be the first in section 3.2.

Answer: Thank you for pointing this out. Moved the sentence above to the beginning of section 3.2 in lines 269-271.

  • The authors should consider presenting the data from section 3.3 in a figure in the main manuscript, not in the supplementary.

Answer: Agree. The data from section 3.3 in Figure 4 of the main manuscript was presented.

  • The authors should relate genes to the proteins and their functions, so their findings would have a more relevant meaning.

Answer: Thank you for pointing this out. Related the genes to the proteins and their functions in lines 483-503.

  • They should add a color scale to explain what the color difference represents in Figure 4.

Answer: Thank you for pointing this out. Figure 4 now has a color scale representing the color difference.

  • If most of the data is in the supplementary, it’s hard to follow what is described in the Results section.

Answer: Agree. Shifted Supplementary Figures S4 and S9 to the main manuscript.

Reviewer 2 Report

Comments and Suggestions for Authors

Dear Editor in Chief,

The authors of the manuscript entitled "Novel Insights into T-Cell Exhaustion and Cancer Biomarkers in PDAC Using scRNA-Seq" employed single-cell RNA sequencing (scRNA-seq) analysis to identify the upregulated genes of T-cells and of cancer cells in two groups (“cancer cells_vs_all-PDAC” and “cancer-PDAC_vs_all-normal”). And after a comprehensive review of the manuscript I have the following comments:

  1. The manuscript is presenting a novel approach for identifying tumor markers in a challenging cancer PDAC in association with the immune cells, specifically T-cell.
  2. The work highlight the importance of tumor microenvironment in understanding the biology of cancer.
  3. They used scRNA-seq gene expression analysis, pathway enrichment PPI and survival analysis to show the outcomes which is very organized methodology.
  4. The results showed and association between gene expression profile and the immune mechanisms in tumor microenvironment by identifying novel genes such as H4C6 and H3C12.
  5. The results are well presented in the provided figures and supplementary materials.

Generally, the manuscript has a scientific merit in the field, however, I have some minor concerns:

  1. The study is based in bioinformatics tools which might need further experimental approaches which limits its translational application, the authors advised to mention this before suggestion of clinical application as mentioned in the conclusion.
  2. I have conservation on the term of novel genes because some genes are well known to be associated with cancers such as TP53. So, it’s better to clarify where are these genes novel, is it tumor or immune.
  3. More discussion is required when talking about the exhaustion of certain T-cells like CD8+ and CD4+.
  4. Some phrases are repeated in many paragraphs such as (this study revealed).
  5. I couldn’t find Kaplan-Meier survival curves.

Author Response

The authors of the manuscript entitled "Novel Insights into T-Cell Exhaustion and Cancer Biomarkers in PDAC Using scRNA-Seq" employed single-cell RNA sequencing (scRNA-seq) analysis to identify the upregulated genes of T-cells and of cancer cells in two groups (“cancer cells_vs_all-PDAC” and “cancer-PDAC_vs_all-normal”). And after a comprehensive review of the manuscript I have the following comments.

  1. The manuscript is presenting a novel approach for identifying tumor markers in a challenging cancer PDAC in association with the immune cells, specifically T-cell.
  2. The work highlight the importance of tumor microenvironment in understanding the biology of cancer.
  3. They used scRNA-seq gene expression analysis, pathway enrichment PPI and survival analysis to show the outcomes which is very organized methodology.
  4. The results showed and association between gene expression profile and the immune mechanisms in tumor microenvironment by identifying novel genes such as H4C6 and H3C12.
  5. The results are well presented in the provided figures and supplementary materials.

Generally, the manuscript has scientific merit in the field, however, I have some minor concerns:

  1. The study is based in bioinformatics tools which might need further experimental approaches which limits its translational application, the authors advised to mention this before suggestion of clinical application as mentioned in the conclusion.

Answer: Thank you for pointing this out. Mentioned the experimental validation of in-silico analysis in lines 658-659 and 665-667.

  1. I have conservation on the term of novel genes because some genes are well known to be associated with cancers such as TP53. So, it’s better to clarify where are these genes novel, is it tumor or immune.

Answer: Indeed, the identified genes in this study are well known to be associated with cancers. However, in this study, these genes are novel in the context of immune cells (T-cell subsets) as they are expressed in T-cell subsets. For example, in lines 618-620, the TP53, MMP9, FN1, and HSP90AB1 genes are already mentioned to be not reported in T-cell subsets, therefore being novel in corresponding T-cell subsets.

  1. More discussion is required when talking about the exhaustion of certain T-cells like CD8+ and CD4+.

Answer: Agree. Briefly discussed the exhaustion of T-cell subsets in lines 634-641.

  1. Some phrases are repeated in many paragraphs such as (this study revealed).

Answer: Thank you for pointing this out. Found 3 repetitions of the phrase “this study revealed” addressed in lines 573 and 576.

  1. I couldn’t find Kaplan-Meier survival curves.

Answer: Thank you for pointing this out. Kaplan-Meier survival curves were in the Supplementary Document; however, they are now shifted in the main manuscript in Figure 7

Round 2

Reviewer 1 Report

Comments and Suggestions for Authors

The authors provide answers to all my concerns and modify the manuscript accordingly.